# Sustaining the T-cell activity in xenografted psoriasis skin

Pernille Kristine Fisker Christensen[1,2]*, Axel Kornerup Hansen[2], Søren Skov[2], Kåre Engkilde[3], Jesper Larsen[4], Maria Helena Høyer-Hansen[1], Janne Koch[1]

1 LEO Pharma A/S, Ballerup, Denmark, 2 Department of Veterinary and Animal Sciences, Faculty of Health and Medical Sciences, University of Copenhagen, Frederiksberg C, Denmark, 3 Amniotics AB, Lund, Sweden, 4 Bioneer A/S, Hørsholm, Denmark

* dvm.pernille.christensen@gmail.com

**Data Availability Statement:** All relevant data are within the paper and its Supporting information files.

**Funding:** The funder provided support in the form of salaries for authors P.K.F.C, M.H.H.-H. and J.K.

## Abstract

Xenografting of psoriasis skin onto immune deficient mice has been widely used to obtain proof-of-principle of new drug candidates. However, the lack of human T-cell activity in the grafts limits the use of the model. Here, we show that xenografting of lesional skin from psoriasis patients onto human IL-2 NOG mice results in increased numbers of human CD3$^+$ cells in the grafts, axillary lymph nodes and blood from human IL-2 NOG mice compared to C.B-17 scid and NOG mice. In addition, disease relevant human cytokine levels were higher in graft lysates and serum from human IL-2 NOG mice. However, the epidermis was lacking and no efficacy of ustekinumab, a human anti-P40 antibody targeting both IL-12 and IL-23, was shown. Thus, despite the sustained T-cell activity, the model needs further investigations and validation to capture more aspects of psoriasis.

## Introduction

The xenograft mouse model of psoriasis has been widely used for the pre-clinical evaluation of new drug candidates [1–3]. However, the human T-cell activity is decreasing in the grafts over time limiting the use of the model for drugs directly targeting T-cell specific pathways [4–7]. The key driver of psoriasis and a highly recognized drug target is the IL-23/IL-17 pathway [8–10]. Dysregulation of this pathway results in keratinocyte hyperproliferation and chemotaxis of various immune cells into the skin which leads to the erythematous scaly plaques characteristic of psoriasis [8, 11]. Therefore, maintaining the activity of this pathway in the xenograft mouse model of psoriasis is highly desired.

Several attempts have been made to optimize the psoriasis xenograft mouse model by co-injection of stimulated autologous human peripheral blood mononuclear cells (PBMCs). However, the effect on the T-cell activity in the grafts has only been sparsely investigated. In one study, T-cells were identified and IL-1β and IFN-γ were expressed in the grafts after repeated co-injections of PBMCs beneath the grafts [7]. In other studies, intradermal injections of PBMCs in symptomless skin from psoriasis patients induced a psoriatic phenotype in the grafts [2, 12, 13]. Of these, only one study investigated the T-cell activity in the grafts [13]. In this study IL-17A and IL-17F were expressed and treatment with an anti-human IL-23

This study was also funded by LEO Pharma and Innovationsfonden (grant number 5189-00097B) awarded to P.K.F.C. The funders had no role in study design, data collection and analysis, decision to publish, or preparation of the manuscript. The specific roles of these authors are articulated in the 'author contributions' section.

**Competing interests:** I have read the journal's policy and the authors of this manuscript have the following competing interests: P.K.F.C. and M.H.H.-H. are former employees of LEO Pharma A/S. J.K is an employee of LEO Pharma A/S. J.L. is an employee of Bioneer A/S. K.E. is an employee of Amniotics AB. A.K.H. declares that he has collaborated with pharmaceutical industry and received funding from this source, as well as he is the owner of a diabetes related patent as described on https://ivh.ku.dk/english/employees/?pure=en/persons/107126. S.S. declares that he has collaborated with the pharmaceutical industry and received funding from this source as described on https://ivh.ku.dk/english/employees/?pure=en/persons/102444. This does not alter our adherence to PLOS ONE policies on sharing data and materials.

receptor mAb reduced the epidermal thickness, inflammatory cell infiltration and CK16 expression score. However, no significant effects of treatment were observed on either IL-17A or IL-17F gene expression.

In this study, we attempted to maintain the T-cell activity, including the IL-23/IL-17 pathway, in the grafts without co-injection of human PBMCs. Recently, more refined immunodeficient mouse strains have been developed including the human IL-2 (hIL-2) NOG mouse [14]. This mouse strain express constant levels of human IL-2 which is important for T-cell maintenance and activation [15]. Thus, we hypothesized that the T-cell activity in the grafts will be sustained to a higher extent in the hIL-2 NOG mice compared to C.B-17 scid and NOG mice due to the presence of human IL-2.

# Materials and methods

## Animals

Female C.B-17 scid (C.B-$Igh$-$1^b$/IcrTac-$Prkdc^{scid}$), NOG (NOD.Cg-$Prkdc^{scid}$ $Il2rg^{tm1Sug}$/JicTac) and hIL-2 NOG (NOD.Cg-$Prkdc^{scid}$ $Il2rg^{tm1Sug}$ Tg(CMV-IL2)4-2Jic/JicTac) mice, 8–12 weeks old, were purchased from Taconic Biosciences, Inc. (Lille Skensved, Denmark and Germantown, NY, USA). The mice were housed in a specific pathogen-free environment (22°C, 40–60% humidity, 12 hour night/day cycle) in the animal facility at LEO Pharma A/S (Ballerup, Denmark). All nesting and enrichment materials were autoclaved prior to use and the mice were provided with $ad$ $libitum$ access to sterile water and autoclaved feed (Altromin 1324, Brogaarden, Denmark). All animal experimental procedures were approved by the Danish Animal Experiments Inspectorate (Permission Number 2013-15-2934-00812) and performed in accordance with relevant guidelines and regulations including the EU Directive 2010/63/EU, the Danish Animal Experimentation Act LBK No 474', the 3Rs (Refinement, Reduction, Replacement) and the PREPARE guidelines [16, 17]. Reporting was performed in line with the ARRIVE guidelines [18]. All surgery was performed under isoflurane anesthesia and prior to surgery local (40μl of 1:1 mixture of Bupivacaine (2.5mg/ml) and Lidocaine (10mg/ml)) and systemic (25 μl Norodyl Vet (5mg/ml)) analgesia were administered. The day after surgery all mice received systemic analgesia (25 μl Norodyl Vet (5mg/ml)). During the studies all efforts were made to minimize suffering and the mice were terminated by cervical dislocation.

## Human specimens

Anonymized human skin samples were obtained from untreated patients with active moderate-severe psoriasis vulgaris. Skin samples were donated upon informed written consent and sampled in accordance with national legislation in the country of origin and the Declaration of Helsinki. The approval of use was authorized by the Macedonian Bureau of Medicinal Products and the Ethics Committee for Clinical Research at Pauls Stradins Clinical University Hospital, Riga, Latvia. From each patient, one keratome biopsy (app. 500 μm thick) was obtained from lesional skin. The keratome biopsies were placed in cold Dulbecco's Modified Eagle Medium (DMEM, Life technologies, California, USA) supplemented with 2.5 μg/ml amphotericin B (Gibco, cat. nr. 15290018), gentamicin (0.02 mg/ml)/amphotericin B (0.5 μg/ml) (Gibco, cat. nr. R01510) and penicillin (400 U/ml)-streptomycin (400 μg/ml) (Thermo Fisher, cat. nr. 15140122) until use. Upon arrival, three 5 mm punch biopsies were taken from each keratome biopsy for analyses.

## Xenografting of human psoriasis skin onto mice

Keratome biopsies were divided into 1.2 x 1.2 cm pieces and placed on ice in Earl's Balanced Salt Solution (EBBS, Gibco) supplemented with 800 U/ml penicillin and 800 µg/ml streptomycin (Gibco). The keratome pieces were surgically transplanted onto the cranio-ventral back area of anesthetized mice, secured with suture (Syneture Caprosyn, P-13 needle; steril 6–0; colorless) and GLUture Topical Tissue Adhesive. The surgical area was covered with Xeroform occlusive petrolatum gauze dressing bandage from the time of surgery until three days after surgery. After removal of the bandages the grafts were kept moist by topical application of basis ointment (LEO Pharma, Ballerup, Denmark). At termination, graft biopsies, blood and axillary lymph nodes were collected.

In a follow-up study, hIL-2 NOG mice engrafted with lesional psoriasis skin were dosed intra-peritoneally (i.p.) with either ustekinumab (Stelara®, Janssen-Cilag, 0.125 mg/mouse), *InVivo*MAb human IgG1, κ isotype control (BioXcell, BE0297, 0.125 mg/mouse), *InVivo*MAb anti-mouse Ly6G/Ly6C (GR-1) antibody (BioXcell, BE0075, 0.1 mg/mouse) or *InVivo*MAb rat IgG2b isotype control (BioXcell, BE0090, 0.1 mg/mouse) on the day of surgery and 7 days after surgery.

## Flow cytometry

Two axillary lymph nodes (left and right) from each mouse were grinded through 40 µm nylon cell strainers (Corning, cat.nr. 431750) into a well. The mesh and well were washed with 3 ml DPBS and the cell suspension was transferred to a 15 ml tube, centrifuged and the pellet was resuspended in 60 µl DPBS. From each sample, 20 µl of lymph node cell suspension was transferred to a 96-well plate for staining and controls. After collection by cardiac puncture, 20 µl EDTA stabilized blood samples were lysed with PharmLyse (BD Biosciences, cat.nr. 555899) for 20 minutes at room temperature, washed and resuspended in DPBS. Fixable Viability Stain 510 (BD Biosciences, cat. 564406) was added to all samples. Subsequently, blood samples were either stained with human markers (CD45 (PE-Cy7, clone HI30, cat. 557748, BD Biosciences), CD3 (PerCP-Cy5.5, clone OKT3, cat. 45–0037, eBiosciences)) or murine markers (CD45 (FITC, clone 30-F11, cat. 553080, BD Biosciences), Ly6G (APC, clone 1A8, cat. 560599, BD Biosciences). Lymph node cell suspensions were stained with human CD45 (PE-Cy7, clone HI30, cat. 557748, BD Biosciences), CD3 (PerCP-Cy5.5, clone OKT3, cat. 45–0037, eBiosciences), CD4 (FITC, clone RPA-T4, cat. 555346, BD Biosciences) and CD8 (PE, clone HIT8a, cat. 555635, BD Biosciences). The human antibodies did not bind to mouse cell surface markers. Automated compensation was performed with UltraComp eBeads (ThermoFisher, cat.nr. 01-2222-41) stained with each of the individual antibodies and relevant isotype controls were applied for all samples. All samples were analyzed using LSR-II with FACSDiva software (BD Biosciences). Subsequently, data were analyzed by the use of FlowJo software version 10 (FlowJo LLC, Oregon, USA).

## Histological and immunohistochemical staining

Skin punch biopsies were placed in 4% formaldehyde for 24 hours at room temperature, paraffin embedded and sectioned. All skin biopsies were stained with Masson's Trichrome (MT) for the evaluation of epidermis. In addition, biopsies were stained by immunohistochemistry (IHC) with either rabbit anti-human CD3 (polyclonal, DAKO, 3 µg/ml), rabbit anti-KU80 (clone C48E7, Cell Signaling, 0.0355 µg/ml), rabbit anti-human IgG/HRP (polyclonal, DAKO, 13 µg/ml) or mouse anti-human ki67 (MIB-1, DAKO, 0.31 µg/ml). Prior to the immunohistochemical stainings, antigen-retrieval was performed over night at 60˚C in BOND Epitope retrieval solution 2 (Leica Biosystems, Nussloch, Germany) followed by incubation in

BondTM Wash Solution (Leica Biosystems, Nussloch, Germany) for 5 minutes at room temperature. Rabbit anti-human CD3, rabbit anti-K80 and mouse anti-human ki67 were detected with BOND Polymere Refine RED Detection (Leica Biosystems, Nussloch, Germany) with Fast Red. BrightVision Poly-AP-Anti Rabbit IgG (Immunologic, Duiven, Netherlands) was used as secondary antibody for rabbit anti-KU80. The rabbit anti-human IgG/HRP were detected with BOND Polymer Refine Detection (Leica Biosystems, Nussloch, Germany). The slides were mounted with DXP mountant (LEO Pharma, Ballerup, Denmark). Relevant blocking, positive, negative and isotype controls were used for all stainings.

## Protein analyses

Snap frozen graft biopsies were lysed with a Precellys 24 tissue homogenizer (Bertin instruments, Montigny-le-Bretonneux, France) in Precellys Soft tissue homogenizing CK14 tubes (Bertin instruments, Montigny-le-Bretonneux, France) with Cell signaling lysis buffer (Cell Signaling Technology, Leiden, The Netherlands) added Complete Mini Protease Inhibitor Cocktail Tablet (Roche diagnostics, Mannheim, Germany), Halt™ phosphate inhibitor cocktail (Thermo Fisher Scientific, Waltham, MA, USA) and Sodium Orthovanadate (New England Biolabs, Ipswich, MA, USA). After mechanical lysing, tissue homogenates were centrifuged (15000 g, 4°C, 15 min) and the protein concentration was determined with a BCA protein assay kit (Pierce Biotechnology, Rockford, IL, USA). Finally, the protein concentration was adjusted to 2 µg/µl. Tissue lysate and serum were analyzed for human IL-6, IL-17A, IL-17C, IL-22, IL 23, IFN-γ, TNF-α, MIP3α, MCP-1 and IgG by the MSD platform (Meso Scale Diagnostics, Rockville, MD, USA) according to the manufacturer´s instructions with standard curves prepared with a matrix relevant for the sample. Importantly, no cross-reactivity to murine proteins were identified in any of the human assays. As the control group for ustekinumab received human IgG1, this group could not be used as control for the analyses of human IgG in serum. Therefore, as the assay did not cross-react to rat IgG2b, a rat IgG2b isotype control group was used as control.

## Statistical analyses

All statistical analyses were performed using GraphPad Prism version 8.1.1 (GraphPad Software, San Diego, California, USA). A block design in which keratome biopsies from all donors were present in all groups were used. One-way or two-way ANOVA and Tukey´s post hoc test was performed to evaluate differences between groups. Data were log-transformed if they failed D´Agostino & Pearson normality test. If log-transformed data failed normality test or if variances were un-equal (Brown-Forsythe test), Kruskal-Wallis test with Dunn´s multiple comparisons test was used. Results are presented as mean ± standard error of mean (SEM) unless otherwise stated and p-values < 0.05 were considered statistically significant. For the statistical analyses of data obtained from the MSD platform, data below lower limit of detection (LLOD) or equal to NaN were replaced by LLOD and data above higher limit of detection (HLOD) were substituted by HLOD.

## Results

### Human IL-2 was present both systemically and in the grafts from hIL-2 NOG mice

To confirm that human IL-2 was present systemically and in skin grafts from hIL-2 NOG mice, the protein level of human IL-2 was evaluated. On day 15 after surgery, the mean serum protein level of human IL-2 in hIL-2 NOG mice was 622.3 ± 463.1 pg/ml, whereas no human

IL-2 was detected in NOG mice. However, the mean serum protein level of human IL-2 in hIL-2 NOG mice was lower on day 43 after surgery (14.2 ± 19.1 pg/ml). As shown in Fig 1a, human IL-2 was also present in the grafts from hIL-2 NOG mice with significantly higher levels identified at both time points compared to keratomes, C.B-17 scid and NOG mice. Interestingly, significantly more human IL-2 was identified in NOG mice compared to keratomes on day 15, but the levels reversed to baseline on day 43. As NOG mice do not express human IL-2, this could indicate that the human T-cells in the grafts transiently produced IL-2.

## Human IL-2 stimulated human T-cell proliferation and migration from the graft into the mouse circulation

After confirming that human IL-2 was present in the grafts, the presence and proliferation of human T-cells were investigated. As shown in Fig 1b, a massive infiltration of human T-cells was observed in the skin grafts from hIL-2 NOG mice compared to C.B-17 scid and NOG mice 15 days after surgery. Similar results were obtained on day 43 (S1 Fig). In addition, ki67 staining indicated T-cell proliferation in dermis in grafts from hIL-2 NOG mice whereas fewer cells stained positive in C.B-17 scid and NOG mice (Fig 1b).

To evaluate if the human T-cells had migrated from the graft into the mouse circulation, axillary lymph nodes were analyzed by flow cytometry according to the gating strategy shown in S2a Fig. Of note, the lymph nodes obtained from hIL-2 NOG mice varied in size, however, in most mice the lymph node on the side of the transplant were larger than the one on the opposite side. The lymph nodes isolated from C.B-17 scid and NOG mice were small and a limited number of cells were isolated from these strains compared to hIL-2 NOG mice. Human CD45$^+$ cells were identified in the lymph nodes from hIL-2 NOG mice at both time points, with fewer cells present on day 43 (Fig 1c and 1d). The majority of the human CD45$^+$ cells were CD3$^+$ and among these both CD4$^+$ and CD8$^+$ cells were identified in hIL-2 NOG mice (S2b–S2g Fig). The percentage and number of CD4$^+$ T-cells were superior to CD8$^+$ T-cells. In addition, a significantly higher percentage of human CD45$^+$ cells were identified in the blood from hIL-2 NOG mice compared to C.B-17 scid and NOG mice on day 43 (S3a and S3b Fig). Most likely due to the systemic presence of human T-cells, the hIL-2 NOG mice developed signs of graft versus host disease (GvHd). This resulted in reduced overall survival of hIL-2 NOG mice from week three due to critical loss of bodyweight and reduced overall well-being (Fig 1e). GvHd development seemed donor dependent and only manifested in three out of six donors. No signs of GvHd were observed in C.B-17 scid and NOG mice.

Taken together, these data indicate that human IL-2 stimulated T-cell proliferation in the grafts and migration of T-cells into the circulation in hIL-2 NOG mice. However, the systemic presence of the human T-cells resulted in signs of GvHd limiting the duration of the model.

## Human T-cell activity was sustained in xenografted hIL-2 NOG mice

We have shown that human T-cells were present and proliferated in hIL-2 NOG mice after engraftment of lesional psoriasis skin. To investigate the T-cell activity, disease relevant human cytokine levels were analyzed in both grafts and serum. The mean protein levels of IL-17A, IL-22, IFN-γ and TNF-α in grafts from hIL-2 NOG mice were higher on day 15 compared to the levels found in C.B-17 scid and NOG mice (Fig 2a, 2c, 2e and 2g). Compared to pre-transplanted skin, the levels of IL-22, TNF-α and IFN-γ were higher in hIL-2 NOG mice, whereas IL-17A reached a comparable level. However, the cytokine levels decreased from day 15 to 43.

Likewise, human IL-17A, IL-22, IFN-γ and TNF-α protein levels were significantly increased in serum from hIL-2 NOG mice (Fig 2b, 2d, 2f and 2h). The mean IFN-γ and TNF-α

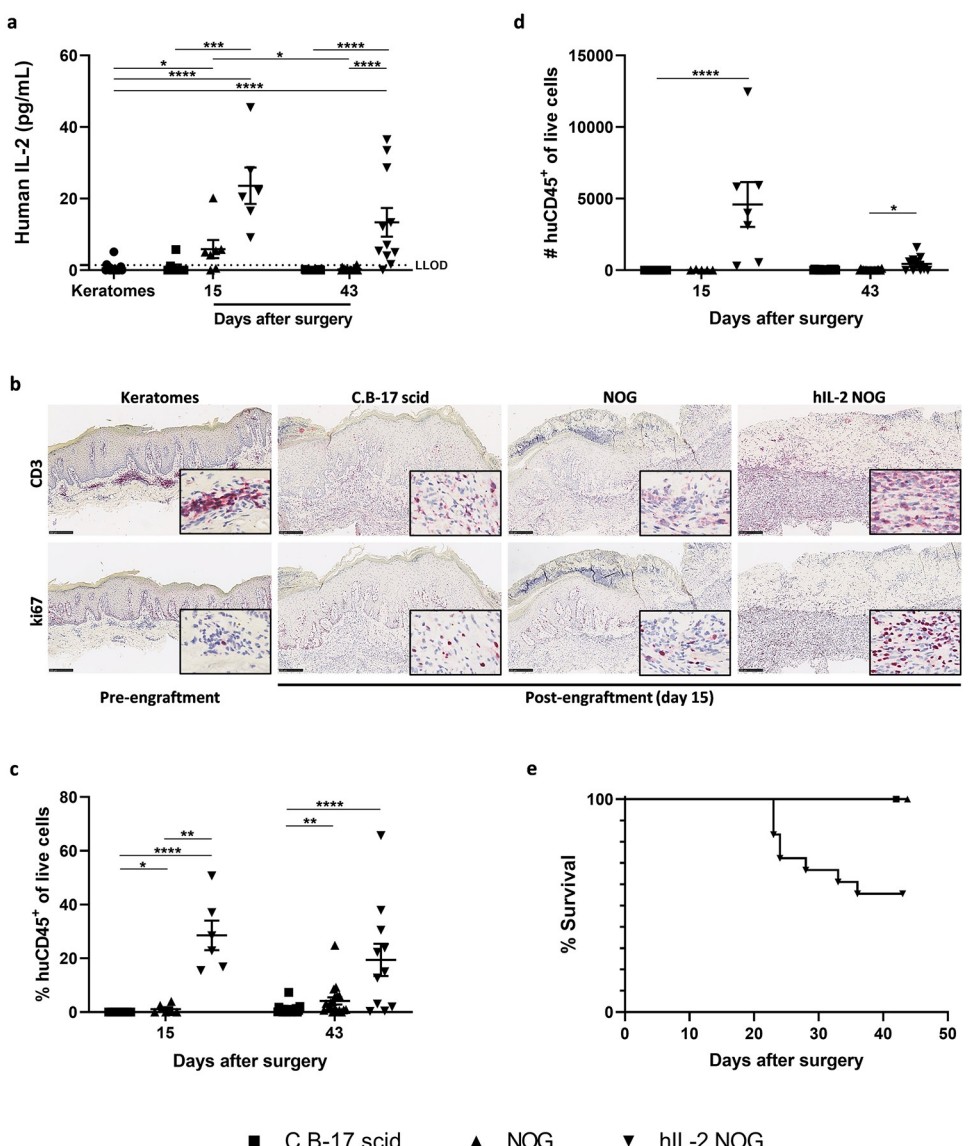

**Fig 1. Human IL-2 protein levels in the grafts, human immune cells in grafts and lymph nodes and survival of hIL-2 NOG mice.** Keratomes from six psoriasis vulgaris patients were included in the study. The number of mice in each group were on day 15: C.B-17 scid n = 8; NOG n = 7; hIL2-NOG n = 6 and on day 43: C.B-17 scid n = 20; NOG n = 20; hIL2-NOG n = 11. (a) Protein levels of human IL-2 were analyzed in keratome (three biopsies from each keratome, n = 18) and graft lysates by the MSD platform (LLOD is lower limit of detection). (b) Immunohistochemical staining with anti-human CD3 and anti-human ki67 on day 15. Representative slides are shown. The bar equals to 250 μm and slides are shown in a 10X magnification with a 20X magnification picture inserted in the lower right corner. (c) Percent and (d) number of human CD45[+] of live cells were analyzed in axillary lymph node cell suspensions by flow cytometry on day 15 and 43. (e) Survival of mice over time after engraftment of psoriasis skin. The survival curves of C.B-17 scid and NOG mice are placed on top of each other, but marked separately (■ and ▲, respectively).

protein levels were maintained from day 15 to 43, however, the IL-17A and IL-22 levels decreased over time. Although not directly comparable, the mean protein level of IFN-γ seemed higher in serum compared to grafts from hIL-2 NOG mice. This could indicate that the inflammation markers in the grafts were lower than in serum, however, it could also be due to degradation of protein during lyzing of the tissue. The IL-17C and IL-23 protein levels

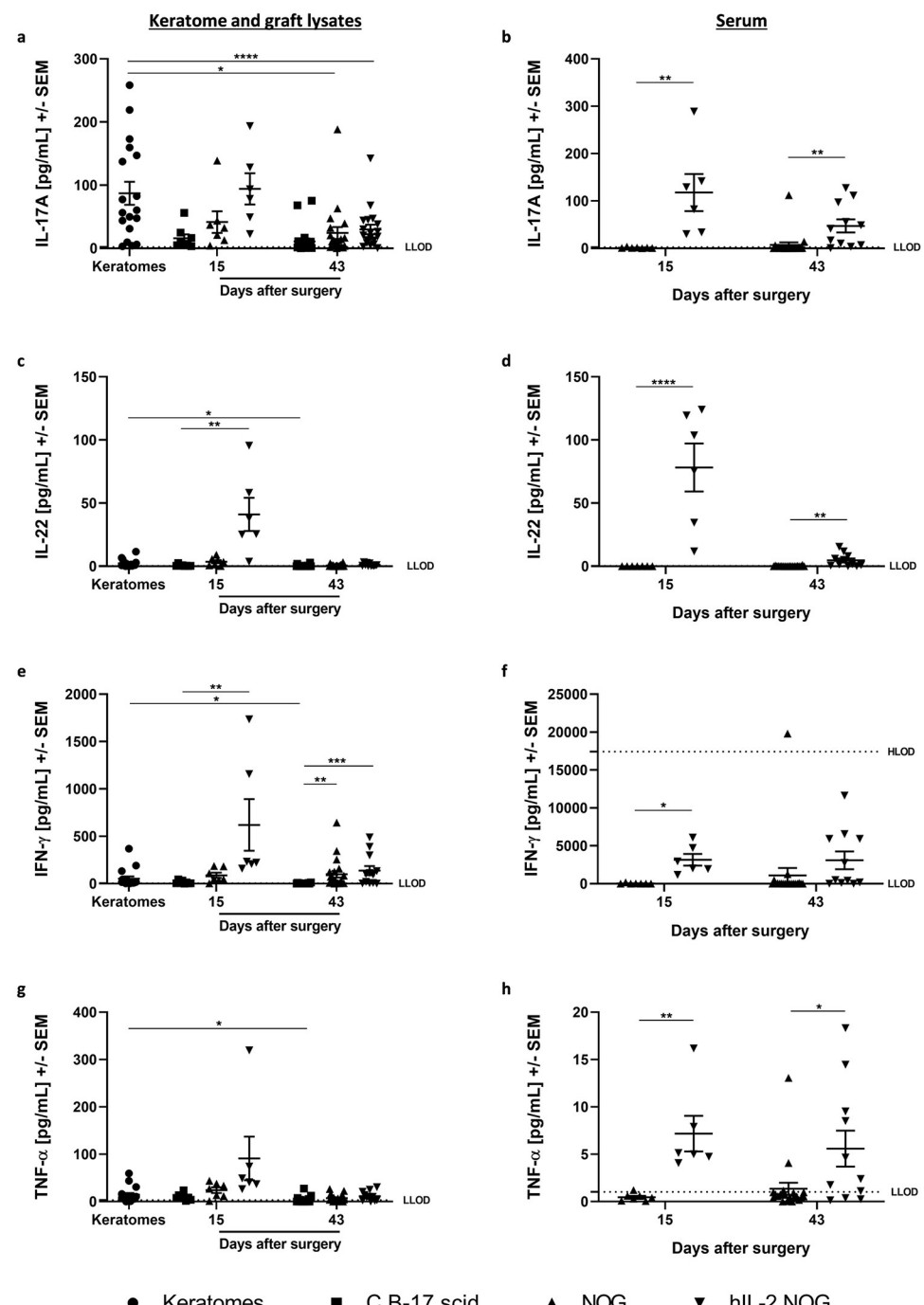

**Fig 2. Disease relevant human cytokine levels in hIL-2 NOG mice compared to C.B-17 scid and NOG mice.**
Human IL-17A, IL-22, IFN-γ and TNF-α protein levels were analyzed in graft lysates (a, c, e, g) and serum (b, d, f, h) on day 15 and 43 by the MSD platform. Keratomes from six psoriasis vulgaris patients were included. The number of mice in each group was on day 15: C.B-17 scid n = 8; NOG n = 7; hIL2-NOG n = 6 and on day 43: C.B-17 scid n = 20; NOG n = 20; hIL2-NOG n = 11. Lysates were generated from biopsies obtained from keratomes at arrival (Three biopsies per keratome, n = 18) and grafts 15 and 43 days after surgery. Serum was isolated from blood obtained by cardiac puncture on day 15 and 43. Human IL-17A (a), IL-22 (c), IFN-γ (e) and TNF-α (g) protein levels in keratomes pre-engraftment and graft lysates from C.B-17 scid, NOG and hIL-2 NOG mice. Human IL-17A (b), IL-22 (d), IFN-γ (f) and TNF-α (h) protein levels in serum from NOG and hIL-2 NOG mice. LLOD is lower limit of detection and HLOD is higher limit of detection.

in grafts were low in all strains at both timepoints and below detection limit in serum (S3c–S3f Fig).

To evaluate the downstream effect of the human cytokines, protein levels of the chemokines MCP-1 and MIP-3α were investigated (S3g and S3h Fig). The MIP-3α protein level in grafts were lower in all mouse strains at both time points compared to fresh keratome biopsies. In contrast, the level of MCP-1 was higher in grafts from C.B-17 scid, NOG and hIL-2 NOG mice 15 days after surgery compared to pre-transplanted skin. The highest mean level was observed in hIL-2 NOG mice, and the levels decreased over time.

In summary, these data indicate that the T-cell activity, identified by the presence of IL-17A, IL-22, IFN-γ and TNF-α, was sustained in hIL-2 NOG mice for at least two weeks. However, the T-cell activity seemed to decrease over time in the grafts.

## The graft size decreases over time and the presence and appearance of epidermis varies after xenografting

After having demonstrated that the T-cell activity was sustained in hIL-2 NOG mice we investigated how this affected the grafts. First of all, the graft size decreased rapidly over time until day 22 in C.B-17 scid, NOG and hIL-2 NOG mice (S4 Fig). From day 22 the graft size seemed to stabilize. As one of the key characteristics of psoriasis is epidermal hyperplasia, the presence and appearance of epidermis were evaluated in all grafts. In the majority of C.B-17 scid mice epidermis maintained a psoriatic appearance on day 15 and 43. However, epidermis was either lacking or partly lacking in NOG and hIL-2 NOG mice on day 15, but fully or partly restored with a psoriatic appearance on day 43 in most mice (Fig 3a). KU80 staining, which only stain human cells, indicated that the restored epidermis was either human, partly human or non-human and confirmed that human cells were present in dermis in most mice across the strains (Fig 3b).

In conclusion, the engraftment of psoriasis skin was superior in C.B-17 scid mice compared to hIL-2 NOG mice. However, the epidermis seemed to be restored in hIL-2 NOG mice in the period between day 15 and 43. Although the duration of the model is limited to approximately three weeks due to GvHd, it could be hypothesized that extending the duration of the model might improve the appearance of epidermis.

## Depletion of murine granulocytes to improve engraftment

It has previously been shown that depletion of murine granulocytes can improve the engraftment of human skin [19]. Thus, engrafted hIL-2 NOG mice were treated with anti-mouse GR1 antibody or isotype control on the day of surgery and seven days after surgery. All mice tolerated the first dose of anti-mouse GR1 antibody, however, immediately after having received the second dose, 6 out of 14 mice became critically ill with signs of severe shock and had to be terminated. Of the surviving mice, an intact epidermis with characteristics of psoriasis was only found in two out of eight mice (S5 Fig). In comparison, no mice from the isotype control group had intact epidermis with characteristics of psoriasis. However, due to the rather high number of mice that had to be euthanized after the granulocyte depletion, no clear conclusion on the effect of granulocyte depletion on graft quality can be drawn.

Taken together, epidermal readouts can become challenging in hIL-2 NOG mice engrafted with psoriasis skin. However, it still might be possible to improve the engraftment without granulocyte depletion by extending the duration of the model providing more time for epidermis to restore.

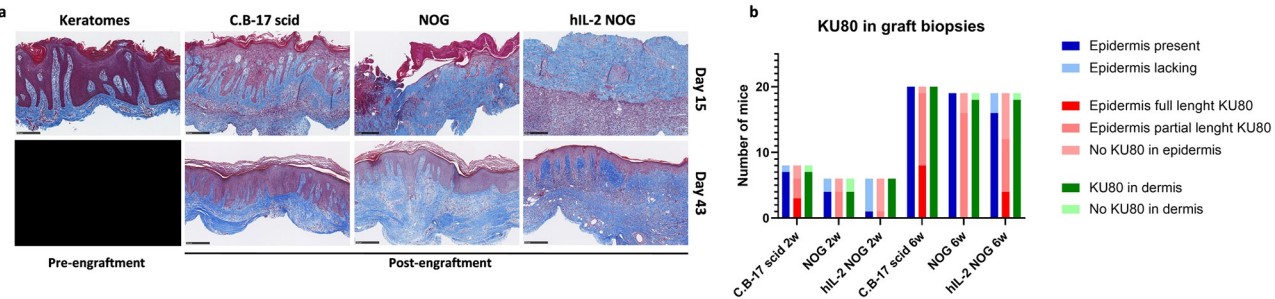

**Fig 3. Presence and appearance of epidermis over time in the different mouse strains.** (a) Masson's Trichrome (MT) stain of keratomes pre-engraftment (six donors, three biopsies from each keratome, n = 18) and graft biopsies from C.B-17 scid, NOG and hIL-2 NOG mice on day 15 (C.B-17 scid n = 8; NOG n = 7; hIL2-NOG n = 6) and 43 (C.B-17 scid n = 20; NOG n = 20; hIL2-NOG n = 11). Representative slides are shown. The bar equals to 250 μm and slides are shown in 10X magnification. (b) The presence of epidermis and KU80 positive stainings in grafts were evaluated. The presence of epidermis was evaluated from MT stainings as either lacking or present (full or partial). Immunohistochemical stainings of the grafts with anti-human KU80 on day 15 and 43 were used to evaluate how much of the epidermis was human (either full length of the biopsy, partial length of the biopsy or no KU80 positive staining in epidermis) and the presence of human cells in dermis (KU80 positive cells present or not present).

## Efficacy of ustekinumab

Due to sustained protein levels of human IL-17A in the grafts from hIL-2 NOG mice 15 days after engraftment, we hypothesized that the IL-23/IL-17 pathway was present. Therefore, we explored the efficacy of ustekinumab, a human IgG1κ mAb targeting the p40 subunit common to IL-12 and IL-23, in the model. However, no differences in human IL-17A, IL-22, IFN-γ and TNF-α protein levels were observed in grafts from ustekinumab treated mice compared to isotype controls (Fig 4a–4d). As previously identified, human IL-23 protein levels were low in both isotype controls and ustekinumab treated mice Fig 4e. In contrast, human IL-6 protein levels were above HLOD in most grafts, but low in keratomes (S6 Fig).

To investigate if the lack of efficacy was due to low exposure of ustekinumab, human IgG protein levels were analyzed in serum. As shown in Fig 4f, significantly more human IgG was detected in serum from ustekinumab treated mice compared to controls treated with rat IgG2b isotype control. Thus, this indicates exposure of ustekinumab in the treated mice.

Interestingly, human IgG protein levels were also detected in the controls. As no cross-reactivity to the rat IgG2b isotype control was observed in the assay, these results indicated that human IgG might be produced in the mice. Thus, the localization of human IgG was evaluated. In keratomes, cells with morphological characteristics resembling plasma cells stained positive for human IgG. Human IgG positive cells were identified in grafts from more than 50% of the mice two weeks after engraftment Fig 4g. In several of the grafts, clusters of human IgG positive cells were identified with additional intercellular matrix staining present, supporting the hypothesis that human IgG were produced and secreted in the grafts. Of note, no human IgG positive cells were observed in grafts from NOG mice.

## Discussion

In the present study, it was shown that the T-cell activity was sustained in grafts after xenografting of lesional human psoriasis skin onto hIL-2 NOG mice. In this model, endogenously produced human IL-2 stimulated the T-cell proliferation and increased the T-cell activity in the grafts. Furthermore, T-cells migrated from the skin into the mouse circulation providing the model with a more systemic disease phenotype. In patients, psoriasis manifests not only in the skin but also systemically with elevated serum levels of disease relevant pro-inflammatory cytokines [20, 21]. Thus, having a systemic component of the disease present in our model

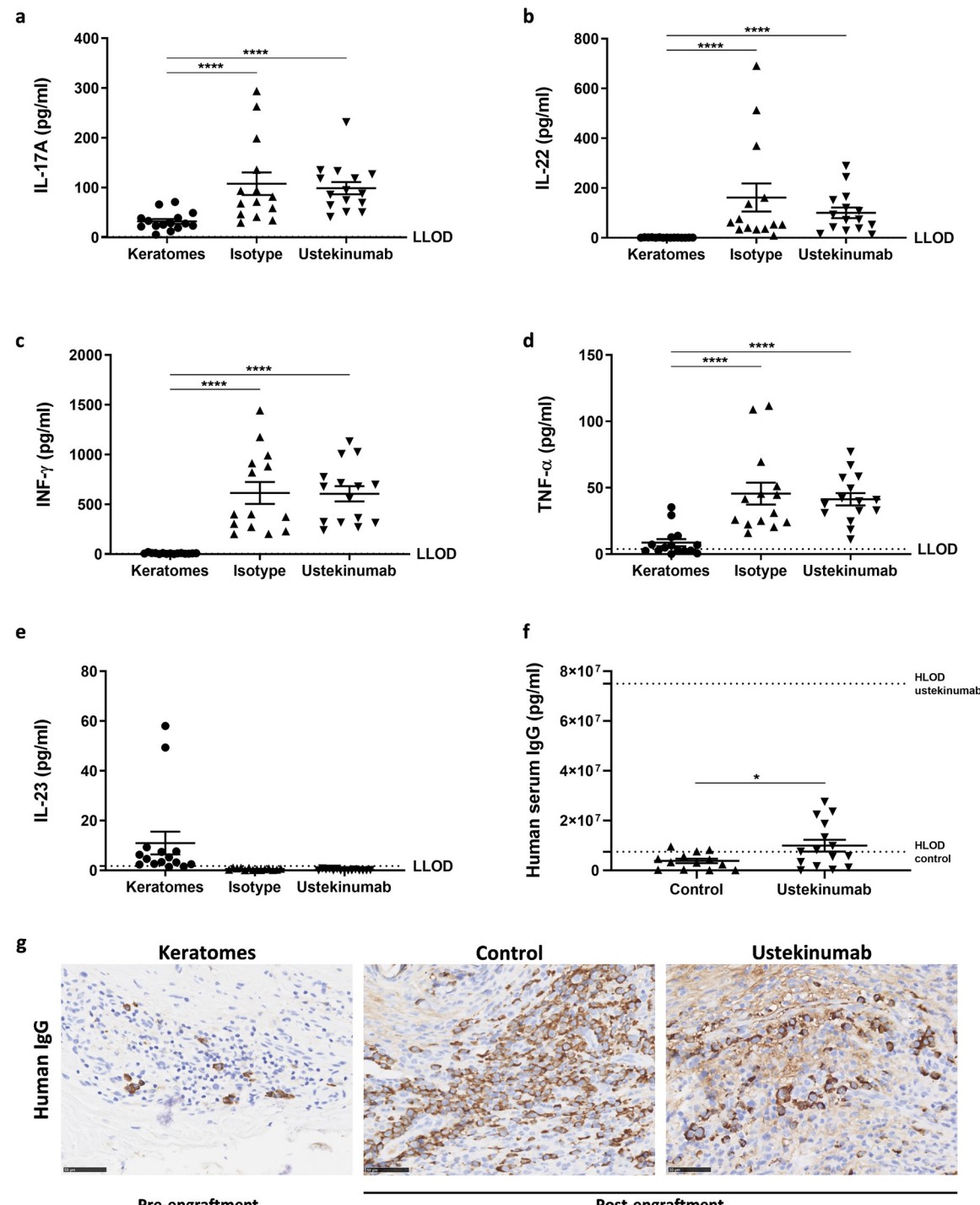

**Fig 4. Effect of ustekinumab and identification of human IgG positive cells in the grafts two weeks post-engraftment.** Protein levels of human IL-17 A (a), IL-22 (b), IFN-γ (c), TNF-α (d) and IL-23 (e) were analysed in keratome (Five psoriasis vulgaris patients were included in the study and three biopsies were obtained per keratome for MSD analyses, n = 15) and graft lysates from hIL-2 NOG mice treated with either ustekinumab or isotype control one week after the final dose. The number of mice were n = 14 (isotype) and n = 15 (ustekinumab). (f) Exposure of ustekinumab was evaluated by analyzing human IgG against an ustekinumab standard curve in serum from ustekinumab treated mice (n = 15) and a control group (n = 13). (g) Immunohistochemical staining with HRP conjugated rabbit anti-human IgG of keratomes pre-engraftment (n = 15) and grafts from controls treated with rat IgG2b isotype (n = 13) and ustekinumab treated mice (n = 15). The HRP conjugated rabbit anti-human IgG did not cross-react to rat. Representative slides are shown. The bar equals to 50 μm and slides are shown in a 20X magnification. LLOD is lower limit of detection and HLOD is higher limit of detection.

could enable the investigation of, not only local, but also the systemic efficacy of new drug candidates. However, the migration of human T-cells into the mouse circulation resulted in GvHd. Both CD4+ and CD8+ T-cells have previously been shown to induce GvHd upon systemic presence in NOG and hIL-2 NOG mice [22]. It was suggested that IL-2 stimulate the differentiation of CD8+ T-cells into memory T-cells which act as key effector cells in severe GvHd. Like psoriasis, GvHd is characterized by increased levels of T-cell derived cytokines such as IFN-γ and IL-17 [22, 23]. This questions if the cytokine levels identified in the model are psoriasis relevant or a product of GvHd. It could be suggested to isolate autoreactive T-cells from the model and engraft these in another mouse xenografted with autologous psoriasis skin to limit the development of GvHd. However, this would be technically challenging and ethically questionable as this would require that the patient had active lesions at both time points, the same patient needed to undergo surgery twice and the patient would not receive treatment for a long period of time.

The protein levels of human IL-17A, IL-22, IFN-γ and TNF-α identified 15 days after transplantation in grafts from hIL-2 NOG mice were comparable to or even higher than the levels in the keratomes before engraftment. It is widely accepted that IL-23 stimulates Th17 cells which produce IL-17A, IL-22, IFN-γ and TNF-α amongst others [24, 25]. In our study, the protein levels of human IL-23 were low in both grafts and serum questioning what stimulated the Th17 cells. Possibly, the low IL-23 protein levels could be an indication of a high use by the Th17 cells. However, treatment with ustekinumab did not support this hypothesis as no effect on relevant human cytokines were observed in the grafts. The lack of efficacy did not seem to be due to lack of exposure as the serum levels of human IgG were significantly higher in ustekinumab treated mice compared to controls. In addition, lower doses of ustekinumab have previously shown effects on epidermal thickness, clinical psoriasis score and keratinocyte proliferation in xenograft mouse models [26, 27]. As ustekinumab does not cross-react with murine IL-23, the lack of efficacy could also lead to the hypothesis that murine IL-23 stimulated the human Th17 cells [28, 29]. However, the hIL-2 NOG mice are severely immunodeficient with functional impairment of their dendritic cells which may compromise their ability to produce murine IL-23 [30]. Finally, the lack of efficacy of ustekinumab could be a consequence of IL-23 not driving the production of human IL-17A. Studies indicate that IL-6 and TGF-β can stimulate the production of IL-17A from T-cells [31, 32]. Hence, the sustained protein level of IL-17A could be dependent of IL-6 and TGF-β in our model. This hypothesis could be supported by the high levels of human IL-6 identified in the grafts. However, further investigations are needed to evaluate this.

Interestingly, high background levels of human IgG in serum seemed to be derived from human IgG producing cells identified in the grafts from hIL-2 NOG mice two weeks after engraftment. It is widely known that IL-2 stimulates T-cell proliferation. However, as human IgG producing cells were not present in grafts from NOG mice but present in grafts from hIL-2 NOG mice, it could be speculated if human IL-2 also supported the proliferation of and immunoglobulin production by these cells. It has previously been shown, that activated B-cells express the IL-2 receptor and that IL-2 stimulates proliferation of B-cells and increases their IgG production in vitro [33]. Therefore, it could be hypothesized that activated human B-cells residing in the psoriasis skin proliferated and increased their IgG production after engraftment onto hIL-2 NOG mice possibly due to stimulation by human IL-2.

One of the key characteristics of psoriasis is increased epidermal thickness. In the present study, an intact epidermis was lacking 15 days after surgery in the majority of the hIL-2 NOG mice. As GvHd and declining cytokine levels limited the duration of the model to approximately 3 weeks, epidermal thickness measurements could become challenging. Depletion of murine granulocytes have previously been shown to improve engraftment in xenograft mouse

models, however, in our attempt this was not feasible due to fatal symptoms of shock. These symptoms have previously been observed in BALB/c and C57BL/6 mice when administering anti-GR1 during inflammation [34–36]. In our study, serum levels of human inflammatory proteins such as TNF-α and IFN-γ indicated systemic inflammation in the surviving engrafted hIL-2 NOG mice. Thus, it could be hypothesized that the systemic inflammation in the mice that did not survive granulocyte depletion were higher than in survivors. This could explain why some mice survived and some did not. However, as the mice had to be terminated immediately no samples were taken and no analyses were performed. Although granulocyte depletion was not feasible in the model, the presence and appearance of the human epidermis could potentially be improved by extending the duration of the model. Until now the characteristics of epidermis has only been explored two weeks after surgery. The model could be extended to three weeks which might provide sufficient time for a better restoration of epidermis. Furthermore, the appearance of epidermis might be improved using another mouse strain. This suggestion is supported by our finding of a superior appearance of epidermis in C.B-17 scid compared to NOG mice at the early timepoint.

In conclusion, xenografting of lesional skin from psoriasis patients onto human IL-2 NOG mice resulted in improved human T-cell activity in the grafts, systemic presence of human T-cells and increased disease relevant human cytokine levels in both grafts and serum compared to C.B-17 SCID and NOG mice. However, the model does not appear suitable for drugs targeting human P40, epidermal readouts are currently not feasible and the model still needs further investigations and validation to more accurately model psoriasis. These investigations should include an evaluation of the contribution of the GvHd to the immune activity, further characterization of the cellular infiltration in the grafts, attempts to improve the appearance of the epidermis and evaluating the efficacy of immune modulating drugs with other targets than P40 such as rizankizumab and infliximab.

## Supporting information

**S1 Fig. Human T-cells and ki67 positive cells in grafts from C.B-17 scid, NOG and hIL-2 NOG mice on day 43.** Immunohistochemical stainings with anti-human CD3 and anti-human ki67. Representative slides are shown. The bar equals to 250 μm and slides are shown in a 10X magnification. Keratomes from six psoriasis vulgaris patients were included in the study. The number of mice in each group were C.B-17 scid n = 20; NOG n = 20; hIL2-NOG n = 11.
(TIF)

**S2 Fig. Gating strategy for flow cytometry on axillary lymph node cell suspensions and human T-cells in lymph nodes on day 43. (a)** Cells of interest was gated on the forward-side scatter. Of these, singlets were identified and live cells were gated from singlets. Next, human CD45$^+$ cells were gated from live cells and from these CD3$^+$ cells were identified. Lastly, CD4$^+$ and CD8$^+$ cells were gated from CD3$^+$ cells. **(b)** percent and **(c)** number of human CD3$^+$ cells of CD45$^+$ cells, **(d)** percent and **(e)** number of human CD4$^+$ cells of CD3$^+$ cells and **(f)** percent and **(g)** number of human CD8$^+$ cells of CD3$^+$ cells in lymph node cell suspensions. Keratomes from six psoriasis vulgaris patients were included in the study. The number of mice in each group were: C.B-17 scid n = 20; NOG n = 19; hIL2-NOG n = 11.
(TIF)

**S3 Fig. Human leukocytes in blood on day 43 and disease relevant human cytokines and chemokines in keratomes, grafts and serum.** Percent **(a)** and number **(b)** of human CD45$^+$ cells of live cells in lysed blood. The gating strategy used was similar to the one used for lymph

node cell suspensions. Human IL-17C and IL-23 were analysed in serum **(c, d)**, keratomes and graft lysates **(e, f)** on day 15 and 43 by the MSD platform. Human MCP-1 **(g)** and MIP-3α **(h)** protein levels in keratomes and graft lysates from C.B-17 scid, NOG and hIL-2 NOG mice. Keratomes from six psoriasis vulgaris patients were included. The number of mice in each group was on day 15: C.B-17 scid n = 8; NOG n = 7; hIL2-NOG n = 6 and on day 43: C.B-17 scid n = 20; NOG n = 20; hIL2-NOG n = 11. Lysates were generated from biopsies obtained from keratomes at arrival (three biopsies per keratome, n = 18) and grafts 15 and 43 days after surgery. Serum was isolated from blood obtained on day 15 and 43. LLOD is lower limit of detection and HLOD is higher limit of detection.
(TIF)

**S4 Fig. Graft size over time.** Graft size was evaluated on day 10, 15, 22, 29 and 36 by measuring the length of two sides of the graft. The baseline (dotted line) is the size of the keratome biopsy transplanted onto the mice on the day of the surgery. All mice were included until day 15. As some mice were terminated on day 15 and pre-maturely, fewer mice were included from day 22–36. C.B-17 scid n = 29 (day 10), n = 29 (day 15), n = 20 (day 22), n = 20 (day 29) and n = 20 (day 36). NOG n = 27 (day 10), n = 26 (day 15), n = 20 (day 22), n = 20 (day 29) and n = 20 (day 36). hIL-2 NOG n = 27 (day 10), n = 27 (day 15), n = 15 (day 22), n = 13 (day 29) and n = 11 (day 36).
(TIF)

**S5 Fig. Graft appearances two weeks after surgery.** Masson's Trichrome staining of grafts from hIL-2 NOG mice treated with either rat IgG2b isotype control (n = 13) or anti-mouse GR1 antibody (n = 8). The bar equals to 500 µm and slides are shown in a 5X magnification.
(TIF)

**S6 Fig. Human IL-6 protein levels in grafts and keratomes.** Protein levels of human IL-6 were analysed in keratome (Five psoriasis vulgaris patients were included in the study and three biopsies were obtained per keratome for MSD analyses, n = 15) and graft lysates from hIL-2 NOG mice treated with either ustekinumab or isotype control. The number of mice were n = 14 (isotype) and n = 15 (ustekinumab). LLOD is lower limit of detection and HLOD is higher limit of detection.
(TIF)

## Acknowledgments

We would like to thank Liselotte Butzkowsky Gurzulidis, Hanne Rosendahl, Maja Wichmann, Dina Silke Malling Damlund, Malene Mandelbaum Beitzel, Stinne Ravnsbæk and Sara Mathez for their technical assistance.

## Author Contributions

**Conceptualization:** Pernille Kristine Fisker Christensen, Axel Kornerup Hansen, Søren Skov, Kåre Engkilde, Jesper Larsen, Maria Helena Høyer-Hansen, Janne Koch.

**Data curation:** Pernille Kristine Fisker Christensen.

**Formal analysis:** Pernille Kristine Fisker Christensen.

**Funding acquisition:** Pernille Kristine Fisker Christensen, Axel Kornerup Hansen, Søren Skov.

**Investigation:** Pernille Kristine Fisker Christensen, Janne Koch.

**Methodology:** Pernille Kristine Fisker Christensen.

**Project administration:** Pernille Kristine Fisker Christensen.

**Supervision:** Axel Kornerup Hansen, Søren Skov, Kåre Engkilde, Jesper Larsen, Maria Helena Høyer-Hansen, Janne Koch.

**Validation:** Axel Kornerup Hansen, Søren Skov, Kåre Engkilde, Jesper Larsen, Maria Helena Høyer-Hansen, Janne Koch.

**Visualization:** Pernille Kristine Fisker Christensen.

**Writing – original draft:** Pernille Kristine Fisker Christensen.

**Writing – review & editing:** Axel Kornerup Hansen, Søren Skov, Kåre Engkilde, Jesper Larsen, Maria Helena Høyer-Hansen, Janne Koch.

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
