## [Decision Letter · Decision Letter 0]

14 Oct 2022

PONE-D-22-24414Sustaining the T-cell activity in xenografted psoriasis skinPLOS ONE

Dear Dr. Christensen,

Thank you for your submission to PLOS ONE. After careful consideration, we feel that it has merit but does not fully meet PLOS ONE’s publication criteria as it currently stands. Therefore, we invite you to submit a revised version of the manuscript that addresses the points raised during the review process. Expert referees have completed their reviews of your team's work and in general have favorable comments about the manuscript. Specific reviewer points regard the content and structure of the paper, which you should adequately address in a revision. The experiments are well-executed and  this work will be a valuable contribution to the literature, however some additional discussion (per reviewer 1) and editing of the prose (both reviewers) will be required. I look forward to a revised version of your manuscript.

We look forward to receiving your revised manuscript.

Sincerely,

Nicholas A. Pullen, Ph.D.

Academic Editor

PLOS ONE

**Journal Requirements:**

"This work was funded by LEO Pharma A/S and the Innovation fund, Denmark (grant number 5189-00097B)."

Please state what role the funders took in the study.  If the funders had no role, please state: ""The funders had no role in study design, data collection and analysis, decision to publish, or preparation of the manuscript."" If this statement is not correct you must amend it as needed. 

"I have read the journal's policy and the authors of this manuscript have the following competing interests: P.K.F.C and M.H.H.-H. are former employees of LEO Pharma A/S. J.K is an employee of LEO Pharma A/S. J.L. is an employee of Bioneer A/S. K.E. is an employee of Amniotics AB. "

We note that one or more of the authors are employed by a commercial company: LEO Pharma A/S., Bioneer A/S., and Amniotics AB.

“The funder provided support in the form of salaries for authors [P.K.F.C, M.H.H.-H. ,J.K, J.L., K.E.], but did not have any additional role in the study design, data collection and analysis, decision to publish, or preparation of the manuscript. The specific roles of these authors are articulated in the ‘author contributions’ section.”

Within your Competing Interests Statement, please confirm that this commercial affiliation does not alter your adherence to all PLOS ONE policies on sharing data and materials by including the following statement: ""This does not alter our adherence to  PLOS ONE policies on sharing data and materials.” (as detailed online in our guide for authors http://journals.plos.org/plosone/s/competing-interests) . If this adherence statement is not accurate and  there are restrictions on sharing of data and/or materials, please state these. Please note that we cannot proceed with consideration of your article until this information has been declared.

Reviewers' comments:

Reviewer's Responses to Questions

**Comments to the Author**

1. Is the manuscript technically sound, and do the data support the conclusions?

Reviewer #1: Yes

Reviewer #2: Partly

2. Has the statistical analysis been performed appropriately and rigorously? 

Reviewer #1: Yes

Reviewer #2: Yes

3. Have the authors made all data underlying the findings in their manuscript fully available?

Reviewer #1: Yes

Reviewer #2: Yes

4. Is the manuscript presented in an intelligible fashion and written in standard English?

Reviewer #1: Yes

Reviewer #2: No

5. Review Comments to the Author

Reviewer #1: Christensen et al. transplanted human psoriasis skin xenografts onto human IL-2 NOG, C.B-17 scid, and NOG mice. The expression of human IL-2 in human IL-2 NOG mice not only sustains T cell activity in the xenografts but also promotes T cells to migrate from the xenografts into the recipient circulation. Therefore, transplantation of human psoriasis skin xenografts induces GVHD in the human IL-2 NOG recipient mice.

The authors also indicate that this model needs further investigations to more accurately model psoriasis.

Comments

The manuscript is well written. The experimental design is elegant, and the experiments are properly executed. The findings are interesting.

Please discuss briefly why T cells in the human psoriasis skin xenografts can induce GVHD in human IL-2 NOG recipient mice. T cells in psoriasis skins may consist of autoimmune, tissue resident, and bystander T cell populations.

In lines 244 and 245, the authors indicate that “the inflammation in the grafts were lower than in serum”. What is serum inflammation? The sentence can be changed into “the inflammation markers in the grafts were lower than in serum”.

Reviewer #2: The authors established a murine xerografted psoriasis model by using human IL-2(hIL-2) NOG mice. Compared with C.B-17 scid and NOG mice, their model showed more T cells in the skin grafts and axillary lymph nodes. Also, higher levels of IL-22, IFN-γ, and TNF-α were observed in the hIL-2 NOG mice model. However, there was lack of efficacy of ustekinumab in terms of inhibition of IL-17A, IL-22, IFN-γ, and TNF-α. Thus, this hIL-1 NOG mrine model of psoriasis needs further investigations to validate its value despite of sustained T-cell activity.

I have some comments and suggestions.

1.Generally speaking, the structure of Results section is not well-organized. Some paragraphs in the Results section are actually figure legends. The description for results in the main text should be different from figure legends. Please separate figure legends (i.e. Line 179, 228, 270, 314, ) from the main text. Also, some figure panels are not described in the main text. For example, the first paragraph of Results section only describes Figure 1d.

2.Some sentences in the manuscript have unclear meaning. The reviewer suggests the authors to sent the manuscript to a native English speaker for English editing.

3.Resolution of figures are too low.

4.The format of references are not unified. Volume and issue numbers are missing in some references. For example, references 1,2,3,7. In addition, it is unnecessary to indicate the pulisher/associations of these journals. For example, in reference 2, no need to mention the Society for Investigative Dermatology, which publishes the Journal of investigative dermatology.

5.Figure 1e, the authors compare survival curves of C.B-17 scid, NOG, and hIL-2 NOG mice. However, only two survial curves are shown. Also, the authors didn't indicate which curves stand for which strains.

6.Line 248, why the authors investigate protein levels of MCP-1 and MIP-3α. What is the significance?

7.Figure 3b, what is the Y-axis? "# of mice"?

6. PLOS authors have the option to publish the peer review history of their article (what does this mean?). If published, this will include your full peer review and any attached files.

Reviewer #1: **Yes: **Wenhao Chen

Reviewer #2: **Yes: **Sebastian Yu

---

## [Author Response · Author response to Decision Letter 0]

8 Nov 2022

Journal Requirements:

• We have to the best of our abilities ensured that the manuscript meets PLOS ONE’s style requirements. 

• Thank you for your comment. The required information has now been included in the methods section (line 65-69).

"This work was funded by LEO Pharma A/S and the Innovation fund, Denmark (grant number 5189-00097B)."

Please state what role the funders took in the study. If the funders had no role, please state: ""The funders had no role in study design, data collection and analysis, decision to publish, or preparation of the manuscript."" If this statement is not correct you must amend it as needed. 

• The amended Role of Funder has now been included in the cover letter. 

"I have read the journal's policy and the authors of this manuscript have the following competing interests: P.K.F.C and M.H.H.-H. are former employees of LEO Pharma A/S. J.K is an employee of LEO Pharma A/S. J.L. is an employee of Bioneer A/S. K.E. is an employee of Amniotics AB. "

We note that one or more of the authors are employed by a commercial company: LEO Pharma A/S., Bioneer A/S., and Amniotics AB.

“The funder provided support in the form of salaries for authors [P.K.F.C, M.H.H.-H. ,J.K, J.L., K.E.], but did not have any additional role in the study design, data collection and analysis, decision to publish, or preparation of the manuscript. The specific roles of these authors are articulated in the ‘author contributions’ section.”

Within your Competing Interests Statement, please confirm that this commercial affiliation does not alter your adherence to all PLOS ONE policies on sharing data and materials by including the following statement: ""This does not alter our adherence to PLOS ONE policies on sharing data and materials.” (as detailed online in our guide for authors http://journals.plos.org/plosone/s/competing-interests). If this adherence statement is not accurate and there are restrictions on sharing of data and/or materials, please state these. Please note that we cannot proceed with consideration of your article until this information has been declared.

• The funding statement and competing interest statement has been updated and included in the cover letter. 

• The reference list has now been corrected and should meet all requirements. 

 

Reviewer 1

Dear Reviewer 1, 

Thank you very much for reviewing our manuscript and for finding our work well-conducted and interesting. We have addressed your comments to the best of our abilities. Please find our answers below. 

Best regards, 

Pernille Christensen

Christensen et al. transplanted human psoriasis skin xenografts onto human IL-2 NOG, C.B-17 scid, and NOG mice. The expression of human IL-2 in human IL-2 NOG mice not only sustains T cell activity in the xenografts but also promotes T cells to migrate from the xenografts into the recipient circulation. Therefore, transplantation of human psoriasis skin xenografts induces GVHD in the human IL-2 NOG recipient mice.

The authors also indicate that this model needs further investigations to more accurately model psoriasis.

Comments

The manuscript is well written. The experimental design is elegant, and the experiments are properly executed. The findings are interesting.

Please discuss briefly why T cells in the human psoriasis skin xenografts can induce GVHD in human IL-2 NOG recipient mice. T cells in psoriasis skins may consist of autoimmune, tissue resident, and bystander T cell populations.

• Thank you for your suggestion. A short discussion of this matter has been included in the revised manuscript (line 362-365). 

In lines 244 and 245, the authors indicate that “the inflammation in the grafts were lower than in serum”. What is serum inflammation? The sentence can be changed into “the inflammation markers in the grafts were lower than in serum”.

• Thank you for making us aware of this unclarity. The sentence has now been changed according to your suggestion (line 255). 

  

Reviewer 2

Dear Reviewer 2, 

Thank you very much for reviewing our manuscript and for your highly relevant comments and suggestions. We have addressed each point to the best of our abilities. Please find our answers below. 

Best regards, 

Pernille Christensen

The authors established a murine xerografted psoriasis model by using human IL-2(hIL-2) NOG mice. Compared with C.B-17 scid and NOG mice, their model showed more T cells in the skin grafts and axillary lymph nodes. Also, higher levels of IL-22, IFN-γ, and TNF-α were observed in the hIL-2 NOG mice model. However, there was lack of efficacy of ustekinumab in terms of inhibition of IL-17A, IL-22, IFN-γ, and TNF-α. Thus, this hIL-1 NOG mrine model of psoriasis needs further investigations to validate its value despite of sustained T-cell activity.

I have some comments and suggestions.

1. Generally speaking, the structure of Results section is not well-organized. Some paragraphs in the Results section are actually figure legends. The description for results in the main text should be different from figure legends. Please separate figure legends (i.e. Line 179, 228, 270, 314, 326) from the main text. Also, some figure panels are not described in the main text. For example, the first paragraph of Results section only describes Figure 1d.

• We are sorry that you find the results section unstructured. We have to the best of our abilities changed the figure legends and paragraphs to separate the figure legends from the main text (line 183, 239, 282, 559). Furthermore, figure 1 has been changed so the first graph mentioned in the text is Fig 1a. 

2.Some sentences in the manuscript have unclear meaning. The reviewer suggests the authors to sent the manuscript to a native English speaker for English editing.

• Thank you for your comment. We have changed some of the sentences and upon acceptance of the manuscript we intent to send it to one of PLOS ONEs language editing partners.

3.Resolution of figures are too low.

• Thank you for your comment. The resolution of all the figures has now been improved. Furthermore, the figures have been uploaded to and accepted by PACE digital diagnostic tool. 

4.The format of references are not unified. Volume and issue numbers are missing in some references. For example, references 1,2,3,7. In addition, it is unnecessary to indicate the pulisher/associations of these journals. For example, in reference 2, no need to mention the Society for Investigative Dermatology, which publishes the Journal of investigative dermatology.

• The references have now been adjusted to make them unified. 

5.Figure 1e, the authors compare survival curves of C.B-17 scid, NOG, and hIL-2 NOG mice. However, only two survial curves are shown. Also, the authors didn't indicate which curves stand for which strains.

• Thank you for noticing this. Two of the survival curves are placed on top of each other which in the graph is marked by both a square and a triangle. This has now been explained in the figure text (line 199-200). 

The identification of the different strains in the graph has been moved from the left side of the figure to the bottom to make it clearer that this description goes for all the graphs in the figure. 

6.Line 248, why the authors investigate protein levels of MCP-1 and MIP-3α. What is the significance?

• Thank you for your highly relevant question. The protein levels of MCP-1 and MIP-3α were investigated to evaluate the downstream effect of the human cytokines. This explanation has now been included in the revised manuscript (line 259-260). 

7.Figure 3b, what is the Y-axis? "# of mice"?

• We are sorry for the confusion. The figure legend has now been changed to “Number of mice”.

---

## [Editor Report · Decision Letter 1]

16 Nov 2022

Sustaining the T-cell activity in xenografted psoriasis skin

PONE-D-22-24414R1

Dear Dr. Christensen,

We’re pleased to inform you that your manuscript has been judged scientifically suitable for publication and will be formally accepted for publication once it meets all outstanding technical requirements.

Kind regards,

Nicholas A. Pullen, Ph.D.

Academic Editor

PLOS ONE

Additional Editor Comments (optional):

Dear Dr. Christensen,

Thank you for submitting your revised manuscript. As the previous reviewers' concerns were all textual and formatting issues, I confirm that you have judiciously addressed their comments. Therefore, I am also pleased to endorse your manuscript for publication. Please note that during production there will likely be additional requests to meet journal format more closely.

I look forward to the eventual release of your paper, congratulations!

Sincerely,

Nicholas A. Pullen, Ph.D.

Editorial Board Member
---

## [Editor Report · Acceptance letter]

3 Jan 2023

PONE-D-22-24414R1 

Sustaining the T-cell activity in xenografted psoriasis skin 

Dear Dr. Christensen:

I'm pleased to inform you that your manuscript has been deemed suitable for publication in PLOS ONE. Congratulations! Your manuscript is now with our production department. 

Kind regards, 

on behalf of

Dr. Nicholas A. Pullen 

Academic Editor

PLOS ONE